# Deep Learning-Based Nystagmus Detection for BPPV Diagnosis

**DOI:** 10.3390/s24113417

**Published:** 2024-05-26

**Authors:** Sae Byeol Mun, Young Jae Kim, Ju Hyoung Lee, Gyu Cheol Han, Sung Ho Cho, Seok Jin, Kwang Gi Kim

**Affiliations:** 1Department of Health Sciences and Technology, Gachon Advanced Institute for Health Sciences and Technology, Gachon University, Incheon 21999, Republic of Korea; 2Gachon Biomedical & Convergence Institute, Gachon University Gil Medical Center, Incheon 21565, Republic of Korea; youngjae@gachon.ac.kr; 3Department of Otolaryngology Head & Neck Surgery, College of Medicine, Gachon University, Incheon 21565, Republic of Korea; febent@gilhospital.com (J.H.L.); han@gilhospital.com (G.C.H.); 4AMJ Co., Ltd., Ansan-si 15610, Republic of Korea; whitemail@naver.com; 5Smith College, Sahmyook University, Seoul 01795, Republic of Korea; 6Department of Biomedical Engineering, College of IT Convergence, Gachon University, Seongnam-si 13120, Republic of Korea

**Keywords:** horizontal nystagmus, convolutional neural network, nystagmus detection, benign paroxysmal positional vertigo, pupil tracking

## Abstract

In this study, we propose a deep learning-based nystagmus detection algorithm using video oculography (VOG) data to diagnose benign paroxysmal positional vertigo (BPPV). Various deep learning architectures were utilized to develop and evaluate nystagmus detection models. Among the four deep learning architectures used in this study, the CNN1D model proposed as a nystagmus detection model demonstrated the best performance, exhibiting a sensitivity of 94.06 ± 0.78%, specificity of 86.39 ± 1.31%, precision of 91.34 ± 0.84%, accuracy of 91.02 ± 0.66%, and an *F*_1_-score of 92.68 ± 0.55%. These results indicate the high accuracy and generalizability of the proposed nystagmus diagnosis algorithm. In conclusion, this study validates the practicality of deep learning in diagnosing BPPV and offers avenues for numerous potential applications of deep learning in the medical diagnostic sector. The findings of this research underscore its importance in enhancing diagnostic accuracy and efficiency in healthcare.

## 1. Introduction

Nystagmus refers to the occurrence of fine eye tremors resulting from challenges in eye movements, often closely associated with peripheral vertigo conditions such as benign paroxysmal positional vertigo (BPPV) [1,2]. This phenomenon significantly impacts visual stability and movement control, with BPPV causing inconvenience and safety issues in daily life due to the eyes’ inability to maintain a fixed position [3,4]. A diagnosis of nystagmus relies on the subjective experiences of the patient, hence introducing variability into its symptoms. This can present challenges, especially in identifying conditions like BPPV, which requires specialized knowledge. Consequently, accurate diagnosis of BPPV can be time-consuming and error-prone [5,6,7].

Recent efforts have been focused on the research of computer-aided diagnosis (CAD) for nystagmus [8]. CAD utilizes image analysis and pattern recognition technologies to swiftly analyze information related to specific patterns of nystagmus. This enables clinicians to make more accurate and prompt diagnoses, thereby improving patient safety and the efficacy of treatments.

In 2021, Zhang et al. developed a deep learning-based model for detecting nystagmus in video recordings of VOG tests. Training and validation were conducted using 8000 data entries, and the proposed torsion-aware bi-stream identification network (TBSIN) model achieved an *F*_1_-score of 0.81 [9]. In 2019, Lim et al. proposed a model for diagnosing benign paroxysmal positional vertigo (BPPV) by detecting the presence of nystagmus in infrared video recordings of VOG tests. Using data from 1005 patients and employing 10-fold cross validation, their LSTM-based fusion model reached an *F*_1_-score of 0.89 [10]. In 2022, Trung Xuan Pham et al. developed a deep learning-based model for detecting nystagmus in video recordings of eye movement tests taken using VOG. The study used 746 samples to diagnose nystagmus in six categories, achieving an *F*_1_-score of 0.90 [11]. In 2023, Haibo Li et al. developed a deep learning-based nystagmus recognition model for vertigo diagnosis using infrared videos obtained from an eye movement recorder. They utilized a two-dimensional convolutional neural network (CNN) combined with Bi-Directional Long Short-Term Memory (BiLSTM), termed CNN-BiLSTM, for training and validation on a dataset comprising 24,521 videos obtained from 1236 patients. They achieved a sensitivity of 91.20% [12]. In 2024, Hang Lu et al. developed a deep learning-based model for diagnosing benign paroxysmal positional vertigo (BPPV) using data from 518 patients who visited the hospital for infrared nystagmus diagnostic tests. They trained and validated their model using the BKTDN (Big Kernel Temporal Difference Network) model, which enhances the TDN (Temporal Difference Module) model with a big kernel long-term module. On average, their model demonstrated an accuracy of 81.7% [13].

However, previous studies still faced several limitations and issues. Firstly, while prior research has investigated the automatic detection of eye oscillations, a comprehensive AI-based BPPV diagnosis system has not yet been implemented. This limitation indicates that manual intervention by medical professionals is still required throughout the process of AI-based nystagmus diagnosis due to the complexity of nystagmus diagnosis. Medical practitioners need to manually analyze vast amounts of data to accurately identify the cause of nystagmus and develop an appropriate treatment plan based on that, leading to an inefficient and time-consuming process. Furthermore, this suggests that the research to date has not succeeded in implementing an optimized process for nystagmus diagnosis, and automatically delivering accurate diagnoses considering the various causes and complex symptoms of nystagmus remains a challenging task.

In this paper, we propose a nystagmus detection model for diagnosing subjects’ BPPV based on deep learning. To build a reliable and robust AI diagnosis system, we developed a nystagmus detection model utilizing the horizontal movement of the pupils extracted through a pupil movement analysis algorithm. The proposed model structure is as follows: The pupils were segmented for movement tracking from video nystagmography data, recorded for BPPV diagnosis. Pupil movement was quantitatively extracted from the segmented pupils using a pupil movement analysis algorithm. Nystagmus was detected using the extracted pupil movement data. We conclude this paper by presenting the experimental results and subsequent analysis of the nystagmus detection model.

## 2. Materials and Methods

### 2.1. Development Environment

The deep learning training system described in this study was equipped with four NVIDIA Tesla V100 graphics processing units (GPUs) (NVIDIA, Santa Clara, CA, USA), 1.2 TB of RAM, and two twenty-core Intel (R) Xeon (R) Gold 5218R processors(Intel, Santa Clara, CA, USA) operating at 2.10 GHz. The training was conducted on an Ubuntu 18.04.5 operating system using Python (version 3.6) and the TensorFlow (version 2.2.0) framework. Additionally, the OpenCV library (version 4.5.1) was used for data loading and preprocessing.

### 2.2. Data Collection

A total of 3363 BPPV test videos were collected from 828 patients who visited Gachon University Gil Hospital for a video nystagmus test between 2 November 2021 and 7 September 2022. These videos were recorded using an ICS Chartr 200 device (Otometrics, a division of Natus, Copenhagen, Denmark) at 30 fps with a resolution of 320 × 120. The lengths of the collected videos varied by up to 5225 frames and were recorded in an infrared environment. All data were collected with the approval of the Gachon University Gil Hospital Institutional Review Board (GBIRB2023-313).

### 2.3. Data Labeling

In this study, our team developed in-house software capable of independently extracting two types of labeling for pupil tracking and nystagmus detection models. An ophthalmologist was trained on the developed software to manually perform the labeling for ground truth (GT) data.

#### 2.3.1. GT Data for Pupil Tracking Model Development

The labeling data used for the development of the pupil tracking model mentioned in this paper consisted of binary images of the pupil area. In the study, 2160 frames were extracted from VOG, and in each frame, the area of the pupils of both eyes was labeled with dimensions of 320 × 120.

#### 2.3.2. GT Data for Nystagmus Detection Model Development

Nystagmus refers to the phenomenon of continuous and repetitive rapid eye movements that originate from slow eye movements and swiftly return to the original position [14]. In the collected VOG data, videos displaying symptoms of nystagmus often do not exhibit these symptoms in the initial frames, but they become more pronounced in the later parts of the video. Therefore, labeling entire videos would be impractical. In this study, frames presenting nystagmus were distinguished on a frame-by-frame basis for all 3363 raw video data. Videos showing continuous and rapid eye movements oscillating from the origin point were labeled as positive.

### 2.4. Data Preprocessing

#### 2.4.1. Data Preprocessing for Pupil Tracking Model Development

To construct the training data, the left and right areas of the eyes shown in the video images were separated and extracted. The extracted images were resampled to dimensions of 320 × 240 and, given the nature of infrared images, were converted into grayscale. Data augmentation was performed on the collected images via horizontal flipping to create mirrored images, while random rotations (≤15°) and horizontal shifts (5%) were used to create a more robust model.

#### 2.4.2. Data Preprocessing for Nystagmus-Detection Model Development

Given the variation in length of up to 5225 frames, all the videos were divided into segments of 1000 frames each. Since videos displaying symptoms of nystagmus often do not exhibit these symptoms in the initial frames, which become more pronounced in the later parts of the video, the frames were deleted in reverse in segments of 1000 frames. Videos with a remaining length shorter than 1000 frames were discarded. The optimal frame length was determined through experimentation to optimize the model’s ability to recognize patterns and generalize. As a result of this preprocessing, the 3363 nystagmus video clips were divided into a total of 27,452 clips.

For the segmentation GT data, each of the 1000 frames of a segment labeled as nystagmus-positive was labeled. For the detection GT data, if more than 1% of the frames out of 1000 contained symptoms of nystagmus, the segment was classified as positive; otherwise, it was marked as negative.

### 2.5. Pupil Tracking and Quantification of Movement

For pupil movement tracking, a CNN-based 2D U-Net model architecture was utilized for pupil segmentation [15]. To train and validate the deep learning model, the data were randomly divided into training, validation, and test sets at an 8:1:1 ratio, each consisting of 6912, 216, and 216 samples, respectively. For each sample in the training set, there were 3 corresponding data points generated through augmentation. Model training was conducted with a learning rate of 1 × 10^−4^, 200 epochs, and a batch size of 16. The outermost coordinates of the segmented pupil area were extracted to determine the center coordinates of the pupil. Using these extracted coordinates, the best-fitting ellipse was calculated using the least-squares fitting algorithm [16,17]. The extracted elliptical data were depicted as shown in Figure 1.

To quantify the pupil movement using the extracted center coordinates, the horizontal movements of the pupil center were converted into 1D time-series data (Figure 2). However, there were cases in which the pupil center was not detected or falsely detected due to closed eyes or inaccurate pupil segmentation. In such cases, the data from the frames before and after the false detection were also likely to be inaccurate. Therefore, the coordinates at which inaccurate detections occurred, along with certain sections around those frames, were designated as inaccurately detected segments. To minimize errors in the extraction algorithm, segments suspected of containing inaccurate detections were treated as missing data.

### 2.6. Segmented Data Correction through a Bridging Algorithm

In patients with nystagmus, the typical eye movement pattern starts with slow eye movements, followed by a return to the original position with quick, saccadic movements, resembling a sawtooth pattern [14]. Given these characteristics, the use of linear interpolation as a bridging algorithm can inadvertently generate nystagmus-like movements, even in the absence of the condition. To address this issue, this study adopts an alternative approach to correcting missing data. This method fills in missing values with the position of the pupil detected immediately before the gap, denoted as NA, thereby avoiding the inadvertent generation of nystagmus-like patterns. This approach is illustrated in Figure 3.

### 2.7. Nystagmus Detection Algorithm

In this study, various network architectures were utilized to train models for video-based nystagmus detection, including a CNN-based GoogLeNet, a Residual Neural Network (ResNet), a custom-developed 1D network model (CNN1D), and a neural network combining CNN1D with a long short-term memory (LSTM) network (CNN-LSTM1D). This led to the development of four different network structures as candidates for the nystagmus diagnosis model. GoogLeNet is known for its deep and complex structure and uses inception modules for efficient feature extraction [18]. ResNet introduces residual blocks to address the vanishing gradient problem encountered while training deep neural networks, allowing for the training of even deeper networks [19].

To detect nystagmus, this study proposes two custom-developed network structures: CNN1D and CNN-LSTM1D. The proposed CNN1D model adopts a shallow CNN-based network structure to prevent data loss and overfitting. The network consists of a total of 14 layers. The input layer has dimensions of 1000 by 1, corresponding to the x-axis movement features of the pupil. Subsequently, two 1D convolution layers with a kernel size of three were used to extract the basic features of the data. A max pooling layer was used to reduce the dimensionality of the feature map and extract strong features, and this process was repeated once more. Two additional 1D convolution layers followed, and Global Average Pooling (GAP) was used before the final output to reduce the number of features. Finally, two Fully Connected (FC) layers were used as the classifier, and batch normalization was performed before the final FC layer. The final output layer followed, with the total number of trainable parameters reaching 92,305.

The CNN-LSTM1D network structure combines CNN1D with an LSTM block to leverage both the local and spatial features of signals, as well as their temporal characteristics [20]. The network comprised 14 layers. Resembling the CNN1D structure closely, the Global Average Pooling (GAP) layer was replaced with a maximum pooling layer. This was followed by an LSTM layer and two FC layers. The final output layer completes the architecture, with the total number of trainable parameters at 117,781.

As previously mentioned, for the training and validation of the deep learning model, the training, validation, and test sets were randomly split at an 8:1:1 ratio based on the number of patients, resulting in 2656, 347, and 360 videos, respectively. These videos were further split into 21,856, 2840, and 2756 clips. The Adam optimizer was used to train each network with a learning rate of 0.00001 and a batch size of 32 [21]. All the networks were trained for over 200 epochs, with the best model extracted for optimization and early stopping applied to prevent overfitting. The mean absolute error (MAE) was selected as the loss function, and accuracy was chosen as the performance metric. All the networks employed a 1D convolutional architecture to process the input data, representing the horizontal movement of the pupils over time. The proposed structures of the two models are depicted in Figure 4.

## 3. Results

### 3.1. Performance Evaluation of the Pupil Segmentation Models for Pupil Tracking

To assess the performance of the trained pupil segmentation model in pupil tracking, its accuracy was evaluated using a randomly selected test set (*N* = 216). The performance evaluation was conducted using a confusion matrix to calculate the true positive (TP), false positive (FP), true negative (TN), and false negative (FN) values. Metrics including sensitivity, specificity, precision, accuracy, and the dice similarity coefficient score (DSC) were measured using the derived confusion matrix [22,23,24,25]. The performance evaluation results for the nystagmus segmentation model exhibited a recall of 97.51 ± 4.40%, precision of 97.98 ± 2.93%, accuracy of 99.92 ± 0.07%, and a DSC of 97.65 ± 3.09% (Table 1).

### 3.2. Performance Evaluation of the Nystagmus Detection Models

The nystagmus detection models were trained using a training set (*N* = 2656) and a validation set (*N* = 347), and their performance was evaluated using a randomly selected test set (*N* = 360). To assess the performance of the nystagmus detection models, metrics including sensitivity, specificity, precision, accuracy, and *F*_1_-score were measured using a confusion matrix. The results exhibited metrics of the highest performance with sensitivity at 95.68 ± 0.79%, specificity at 86.94 ± 1.42%, precision at 91.53 ± 0.96%, accuracy at 91.02 ± 0.66%, and an *F*_1_-score of 92.68 ± 0.55% (Table 2).

A comparison of the receiver operating characteristic (ROC) curves of the nystagmus detection models can be seen in Figure 5 [23]. The four models achieved high area under the curve (AUC) ROC values ranging between 0.83 and 0.90. Analysis of the ROC curves indicated that all the classifiers performed adequately, with statistically significant differences in the AUC performance among the four models. Notably, the CNN1D model was identified as the most suitable model for nystagmus detection, exhibiting the largest AUC of 0.902. This indicates that the CNN1D model can achieve higher accuracy in detecting nystagmus compared to the other models.

## 4. Discussion

In this study, we propose a deep learning-based nystagmus detection algorithm for the diagnosis of BPPV using VOG data. To predict horizontal nystagmus, 1D pupil movement time-series data were extracted from the VOG data using U-Net2D. Based on the extracted data, four types of architectures were referenced to train the nystagmus detection models: GoogLeNet1D, ResNet1D, and two custom-developed models, CNN1D and CNN-LSTM1D.

Table 3 compares the performance of the models developed in this study with those from other studies that utilized machine learning- and deep learning-based nystagmus detection. All the models were designed to diagnose nystagmus from infrared-based nystagmus diagnostic video footage, with the methods or techniques listed by study. The results in each study were compared using sensitivity, precision, and *F*_1_-score. Since the reported ranges for these metrics in each study varied from 0 to 1 or from 0 to 100, all the values were adjusted to fall within the range of 0 to 100.

The algorithm proposed in this study for the diagnosis of BPPV demonstrated higher sensitivity and *F*_1_-score values compared to other studies, despite the N value being not as high as other studies, indicating that the 1D model structure contributed to an overall improvement in performance. The results demonstrate that the proposed algorithm is an effective tool for diagnosing BPPV. Moreover, comparisons to other studies emphasize the generalizability and reliability of the methods used in this study for diagnosing BPPV. Considering the complexity of the models used in other studies, the CNN1D model developed in this study provided higher accuracy and sensitivity despite its simplicity, indicating that simplicity can sometimes lead to better performance in specific diagnostic applications.

A comparison of the nystagmus diagnostic performance of the models trained using the four architectures revealed that the CNN1D model exhibited the best performance. This can be attributed to the fact that the training data primarily consisted of 1D time-series data on the horizontal movement of pupils, limiting the amount of information conveyed, likely leading to overfitting in more complex models. By employing a shallower CNN model, it was possible to extract more accurate features from limited information, which helped reduce the complexity of the model while preventing overfitting and improving generalization. These findings underscore the importance of innovation of the algorithm selection and model structure in nystagmus diagnosis, suggesting that future research should explore the application of various data types and model architectures to further enhance nystagmus diagnosis model performance.

However, there are limitations to this study. Analysis of the data that each model failed to correctly diagnose revealed that many instances involved transitions from normal eye responses to nystagmus-like movements. This suggests that while the clear sawtooth pattern of nystagmus did not emerge in the 1D time-series data representing pupil movement, precursor signs of nystagmus were present in the actual VOG data. This phenomenon indicates that detecting nystagmus based solely on 1D time-series pupil movement data presents challenges, particularly in segments where transitions from normal eye movements to nystagmus occur. This difficulty can be attributed to the limitations of single-variable models, which may not accurately capture specific changes in the data. Additionally, there is a need for additional data collection and construction. Although the model’s learning process and results were excellent even at low N values, there is a need to collect and build additional data to reflect this in various clinical environments.

Therefore, to overcome these limitations, future research should explore methods that incorporate not only information on horizontal pupil movement but also other variables, approaching the problem with multivariate data. Integrating a range of variables into training models is expected to enable more accurate differentiation of nystagmus. This multivariate approach is anticipated to be more effective than using short segments in predicting nystagmus, potentially overcoming the limitations of existing models and enhancing the accuracy of nystagmus diagnosis. This suggests the need for further exploration of how different types of data inputs and model architectures can improve the detection and diagnosis of nystagmus, indicating a promising direction for future research in this field. Additionally, there is a need to collect and build additional data so that they can be applied in various environments. Data obtained from a variety of situations and conditions will help improve the accuracy of the model and better detect subtle differences between normal eye movements and nystagmus. This will allow the model to perform more reliably in a variety of clinical scenarios.

## 5. Conclusions

In this paper, we propose a reliable method for nystagmus detection and a new approach to the diagnosis of benign paroxysmal positional vertigo (BPPV) through the integration of deep learning into pupillary analysis. The experimental results demonstrate that by utilizing a pupillary tracking model to detect the pupil area in video frames and classifying the acquired pupillary movement time-series data, nystagmus can be effectively detected. The proposed method has a significantly improved diagnostic accuracy compared to existing methods. The application of deep learning techniques to BPPV diagnosis is crucial for identifying and analyzing complex nystagmus patterns, while pupillary analysis aids in tracking eye movements to yield precise results. This approach expands the scope of deep learning applications in medical diagnostics, enabling faster and more accurate diagnoses, thereby reducing patients’ treatment time, and enhancing the overall quality of medical services. We anticipate meaningful contributions to the field of medicine through these advancements.

## Figures and Tables

**Figure 1 sensors-24-03417-f001:**
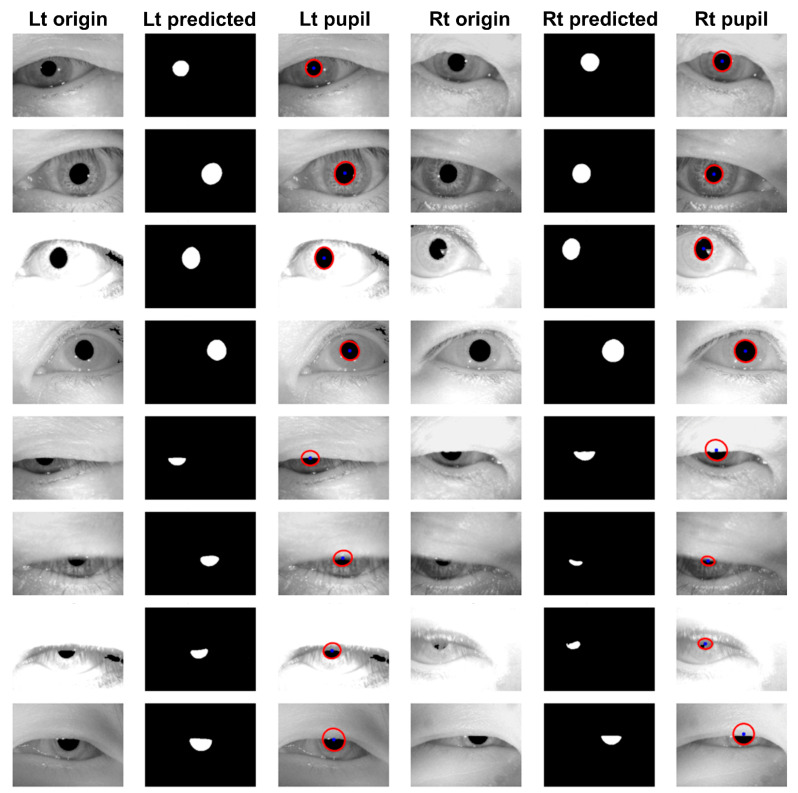
Pupil detection results using a least-squares fitting algorithm. The red circle represents the edge of the pupil, while the blue dot indicates the center of the pupil.

**Figure 2 sensors-24-03417-f002:**
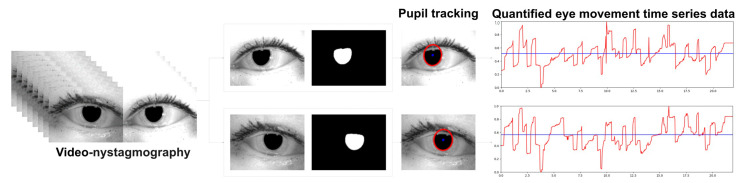
Pupil tracking and eye movement quantification results.

**Figure 3 sensors-24-03417-f003:**
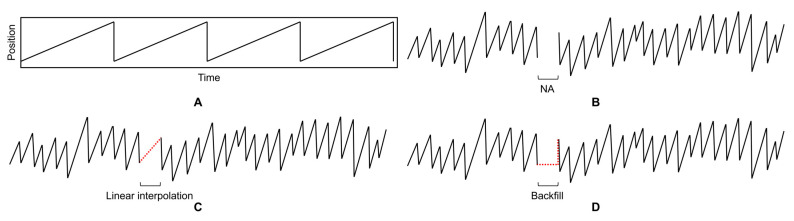
Correcting for missing values (NA) in horizontal pupil movement. (**A**) Serrated eye movements in patients with nystagmus, (**B**) eye movement data with missing values, (**C**) data with linear interpolation, (**D**) data calibrated to the last detected position.

**Figure 4 sensors-24-03417-f004:**
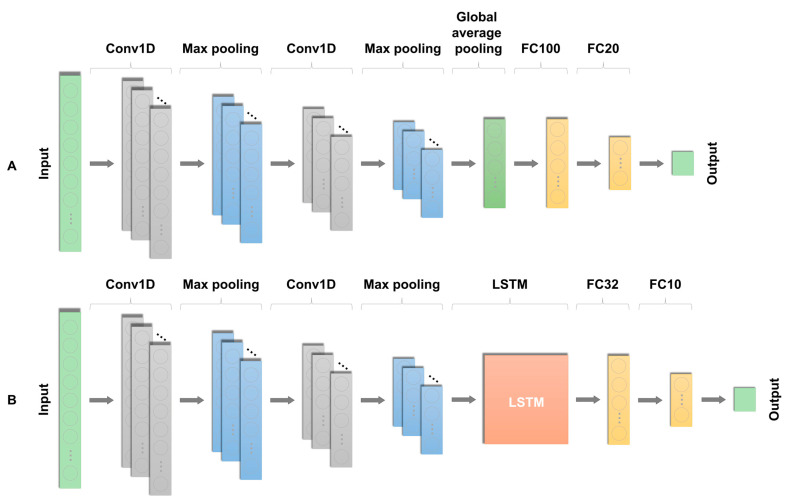
The deep learning model architecture for nystagmus detection. (**A**) The details of the first proposed CNN1D model, (**B**) the details of the second proposed CNN-LSTM1D model.

**Figure 5 sensors-24-03417-f005:**
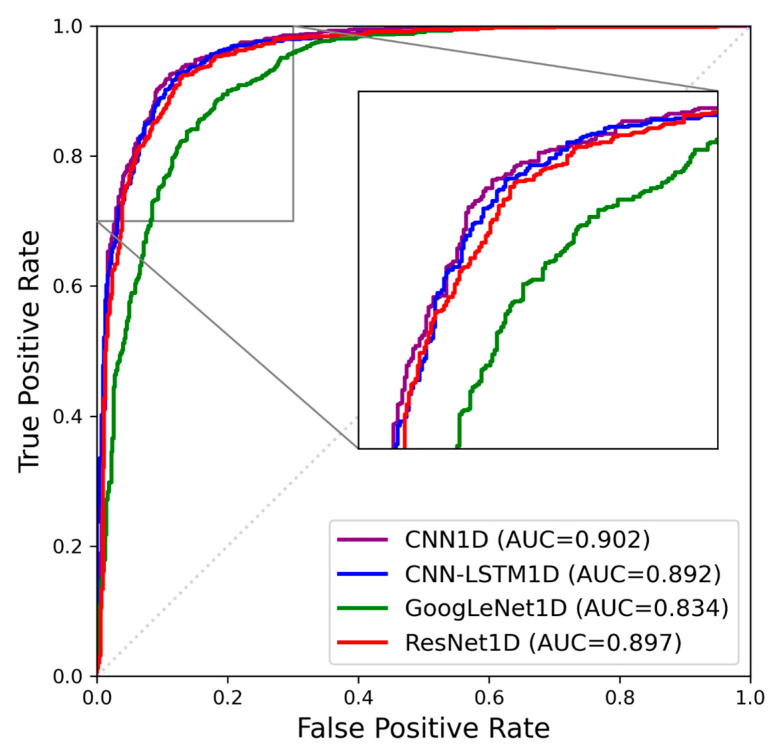
Comparison of ROC curves among the four nystagmus detection models.

**Table 1 sensors-24-03417-t001:** Performance evaluation results of the pupil segmentation model.

	Sensitivity (%)	Specificity (%)	Precision (%)	Accuracy (%)	DSC (%)
U-Net	95.67 ± 9.60	99.96 ± 0.03	97.18 ± 8.12	99.91 ± 0.05	96.29 ± 8.44

**Table 2 sensors-24-03417-t002:** Performance evaluation results of the nystagmus detection models.

	Sensitivity (%)	Specificity (%)	Precision (%)	Accuracy (%)	*F*_1_-Score (%)
CNN1D	94.06 ± 0.78	86.39 ± 1.31	91.34 ± 0.84	91.02 ± 0.66	92.68 ± 0.55
CNN-LSTM1D	95.68 ± 0.79	82.70 ± 1.41	89.40 ± 0.95	90.54 ± 0.76	92.43 ± 0.63
GoogLeNet1D	95.04 ± 0.77	71.76 ± 1.84	83.69 ± 1.20	85.82 ± 0.88	89.00 ± 0.72
ResNet1D	92.56 ± 0.92	86.94 ± 1.42	91.53 ± 0.96	90.33 ± 0.82	92.04 ± 0.69

**Table 3 sensors-24-03417-t003:** Comparison of nystagmus detection achieved in this study with related researchers’ work. (DIM: dimensions, SEN: sensitivity, PRE: precision, *F*_1_: *F*_1_-score).

	Method		DIM	SEN (%)	PRE (%)	*F*_1_ (%)	*N*
Adopted Method	CNN1D	DL	1D	94.06	91.34	92.68	360
Anh et al. (2022) [26]	SVM	ML	-	79.00	78.00	78.00	253
Lim et al. (2019) [10]	CNN	DL	2D	80.80	79.80	79.40	1005
Zhang et al. (2021) [9]	TBSIN	DL	2D	78.92	81.88	81.00	1600
Rham et al. (2022) [27]	SVM	ML	1D	90.00	91.00	90.00	746
Li et al. (2023) [12]	CNN-BiLSTM	DL	2D	91.20	94.30	-	4904

## Data Availability

The raw data supporting the conclusions of this article will be made available by the authors on request.

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
