# Peer review of "Deep Learning-Based Nystagmus Detection for BPPV Diagnosis"

_sensors, 2024, doi:10.3390/s24113417_

Round 1

Reviewer 1 Report

Comments and Suggestions for Authors

This paper proposed a nystagmus detection method using deep learning. The authors have conducted some research on network architecture, but I think there are many details that were not clearly expressed in the paper. First and foremost, this paper mainly proposes a deep learning-based method for detecting nystagmus, but in the data labeling and preprocessing sections, the authors mainly describe pupil detection. What are the criteria for judging nystagmus? How to label for the nystagmus detection model? What are the labels for nystagmus? These are the most relevant issues that are not mentioned in the paper. Moreover, when dealing with missing data in section 2.6, there is no clear statements on how to correct it. Especially Figure 3(D), it is necessary to explain how backfill or calibrate the data. In the introduction, the authors stated that this study aims to overcome the mentioned limitations by implementing a comprehensive AI-based BPPV diagnosis system, I think it would be better to add the contributions of this paper at the end of the introduction, in order to describe the method used to address each limitation separately.

Comments on the Quality of English Language

Minor editing of English language required

Reviewer 2 Report

Comments and Suggestions for Authors

The performance indicators used are not very suitable; As a result, it's best to add a walking path.

Comments on the Quality of English Language

The syntax can be further validated.

Reviewer 3 Report

Comments and Suggestions for Authors

Abstract is good, clear and identifies the methodology and outcomes in a consistent way. The keywords could be reviewed for more specific terms - in the present form they are too general 

Introduction is very good in terms of clearness and coherence, easy to follow and provides the background for the research and experiment. However the existing research could be extended to more sources, please give some more examples of the latest investigations on the matter. 

Section 2 is good divided into subsections for each aspect which makes it easy to follow the experiment. This part of the research is the major contribution and is presented in a very efficient way. All the aspects of the experiment go in a consistent manner which also provides the reliability of the results. 

Results section is brief and clear with 2 tables with the same label. The subsections are titled clearly - however it would be better to label the tables with a mark referring to the title of the subsection. Please check the scale of figure 5 - it looks a little odd.In general the section is very good and clear. 

The results are further developed in the discussion section with the table 3 that sums up the experiment's outcomes.  The section provides the detailed description of the contribution of the reviewed research in comparison with other experiments and in comparison with the models used in the paper. The description is good and thorough, the limitation section would benefit the paper though the authors approached this and of they develop it a bit that would look good especially with such type of data and experiment. 

The conclusion section is rather short even though most of the covered material was given in the previous section.can be left lie this however it seems like the "bullet-pointed" short concluded aspects will make it complete. This is just recommendation.

It was a pleasure reading this  paper, very clear and consistent experiment and methodology. The area is of a great importance and hope the authors will develop their work in a way that the results could find a wide practical application in real world and benefit the society.

Round 2

Reviewer 1 Report

Comments and Suggestions for Authors

The author did not make comprehensive revisions according to my previous review comments. From the perspective of expression, I think the innovation of this paper is not outstanding. And what are the limitations of the proposed method?

Comments on the Quality of English Language

None

Reviewer 3 Report

Comments and Suggestions for Authors

I am ok with the changes introduced and recommend the paper fr publication

Author Response

We would like to thank you once again for your thorough review. it was a pleasure working on your feedback